# Risk Factors of Aggressive Clinical Presentation in Patients with Angiographically Aggressive Cranial Dural Arteriovenous Fistulas

**DOI:** 10.3390/jcm10245835

**Published:** 2021-12-13

**Authors:** Hung-Yu Wen, Hsien-Chung Chen, Shun-Tai Yang

**Affiliations:** 1Division of Neurosurgery, Department of Surgery, Shuang Ho Hospital, Taipei Medical University, New Taipei City 235041, Taiwan; wantedweng@gmail.com (H.-Y.W.); yangcyaa@gmail.com (H.-C.C.); 2Taipei Neuroscience Institute, Taipei Medical University, New Taipei City 235041, Taiwan; 3Department of Surgery, Taoyuan Armed Forces General Hospital, Taoyuan City 325208, Taiwan; 4Graduate Institute of Medical Sciences, College of Medicine, Taipei Medical University, Taipei 110301, Taiwan; 5Department of Surgery, School of Medicine, College of Medicine, Taipei Medical University, Taipei 110301, Taiwan; 6Comprehensive Cancer Center, Taipei Medical University, Taipei 110301, Taiwan

**Keywords:** aggressive cranial dural arteriovenous fistula, venous hypertension, leptomeningeal venous drainage

## Abstract

Compared to nonaggressive cranial dural arteriovenous fistulae (cDAVF), aggressive cDAVF carries leptomeningeal venous drainage (LVD) and has approximately 15% annual risk of hemorrhagic and non-hemorrhagic aggressive neurological presentations. In terms of aggressive clinical presentations, the previous classification does not adequately differentiate the higher risk group from the lower risk group. Herein, we retrospectively collected a series of patients with aggressive cDAVF and explored the risk factors for differentiating the higher-risk group from the lower-risk group with aggressive clinical presentations. We retrospectively collected patients with aggressive cDAVF from March 2011 to March 2019. The risk of aggressive clinical presentation was recorded. Risk factors were included in the analysis for aggressive clinical presentations. From March 2011 to March 2019, 37 patients had aggressive cDAVF. Among them, 24 presented with aggressive clinical presentation (20, hemorrhagic presentation; four, non-hemorrhagic presentation). In patients presenting with hemorrhage, four patients experienced early rebleeding after diagnosis. In the univariate analysis, risk location, directness of LVD, exclusiveness of LVD, and venous strain were significantly different in patients with aggressive clinical presentation. In the multivariate analysis, exclusiveness of LVD and venous strain were observed, with a significant difference between patients with aggressive clinical presentation and those with benign clinical presentation. Among patients with angiographically aggressive cDAVFs, approximately 65% presented with aggressive clinical presentations in our series. Among all potential risk factors, patients with exclusiveness of LVD and venous strain have even higher risk and should be treated aggressively and urgently.

## 1. Introduction

Intracranial dural arteriovenous fistulae (DAVF) are pathological shunts observed between dural arteries and dural venous sinuses, meningeal veins, or cortical veins. Traditional grading systems, including the Cognard and Borden classifications, were used to predict the clinical outcomes of cranial dural arteriovenous shunts and categorize them as aggressive or benign.

Patients with DAVFs with cortical venous regurgitation are stratified as patients with aggressive DAVFs. The annual risks for hemorrhagic and non-hemorrhagic neurological deficits during follow-up in patients with aggressive DAVF were 8.1% and 6.9%, respectively [1], suggesting that most of the patients with DAVFs with cortical venous regurgitation still have relatively benign clinical presentations. However, Duffau et al. reported that 35% of early rebleeding risk occurred within 2 weeks after the first hemorrhage in patients with DAVFs with retrograde cortical venous drainage [2]. This suggests that some patients with DAVF with cortical vein regurgitation carry life-threatening risks, and constant attention and urgent treatment is necessary for these patients.

The Cognard and Borden classifications stratify patients with DAVFs mainly according to the existence of cortical venous regurgitation. However, using this classification system seems to insufficiently identify a variable range of clinical behaviors in patients with angiographically aggressive DAVFs (Borden II and III, Cognard type IIa+b to IV) [3]. Herein, we retrospectively reviewed the clinical characteristics of patients with DAVFs with cortical venous regurgitation at our institution and aimed to explore the potential risk factors of aggressive clinical presentations in such patients. This study highlights the importance of identifying patients with DAVFs at high risk of demonstrating aggressive clinical presentations with early deterioration, and urgent management of such patients to treat or downgrade DAVFs is strongly recommended.

## 2. Materials and Methods

From a series of 37 consecutive patients with aggressive DAVFs based on the Cognard and Borden classifications treated in our department between March 2011 and March 2019, patients with complete angiographic images were retrospectively analyzed. Patients with pediatric and/or congenital DAVF (CDAVF) were excluded from the study. Potential risk factors, such as sex, age, Borden classification, Cognard classification, risky location, directness of leptomeningeal venous drainage (LVD), exclusiveness of LVD, and venous strain, were included in the analysis of aggressive clinical presentations. Among these patients, hemorrhage, focal neurological deficit, increased intracranial pressure (IICP), seizures, altered mental status, and ascending myelopathy were considered aggressive clinical presentations. Tinnitus/bruit, mild-to-moderate headache, ocular symptoms without IICP, and asymptomatic were considered nonaggressive clinical presentations. Fistulas at tentorial area, anterior cranial fossa, foramen magnum, superior sagittal sinus, and petrous venous drainage were considered risky locations [4,5,6]. DAVFs located at the cavernous sinus, lateral sinus, and torcula were stratified as non-risky locations. All patients were analyzed according to the Directness and Exclusivity of LVD and Venous Strain (DES) concept proposed by Baltsavias et al. [7] (Figure 1). Direct LVD was defined as direct venous drainage to the bridging and leptomeningeal veins instead of the venous sinus. An exclusive LVD was defined as venous drainage via only the leptomeningeal veins with the shunt located in a bridging vein, with its exit in the occluded sinus, or the only exit of the sinus was through the bridging vein to the leptomeningeal venous system. Venous decompensation, morphologically expressed as ectatic or congested veins (also known as pseudophlebitic appearance) [8], or both were recorded as cortical venous strains.

### Statistical Method

We performed a univariate analysis first and backward stepwise analysis to further select the significant predictive variables. The selected significant predictive variables were used for the final multivariate analysis for aggressive clinical presentations. For the univariate analysis, the chi-squared and Student’s *t*-tests were used. Logistic regression was used for the multivariate analysis. All analyses were performed using the commercially available Stata version 12.0 software (Stata, College Station, TX, USA). *p* < 0.05 was considered statistically significant.

## 3. Results

Demographics of the recruited patients with angiographically aggressive DVAF are presented in Table 1.

The mean age of all patients at presentation was 60 (range: 27–86) years. Among them, 22 (59%) were men and 15 (41%) were women.

### 3.1. Clinical Presentation and Dural Arteriovenous Fistula Classification

Aggressive clinical presentation was observed in 24 patients (65%), and 20 patients presented with hemorrhage (54%). Moreover, 20 and 17 of all aggressive DAVF cases were Borden types II and type III, respectively. In the Borden classification group, aggressive presentation was observed in 11 and 13 patients with Borden types II and III, respectively. Furthermore, 21 patients had Cognard type IIa + b DAVF, and 11 of them had aggressive presentation. Ten patients with DAVF had Cognard type III DAVF, and aggressive presentation was noted in eight of these patients. The remaining six patients had Cognard type IV DAVF, and five of them had an aggressive presentation. DAVF occurred at a risky location (anterior cranial fossa, tentorium, foramen magnum, and superior sagittal sinus) in 21 patients. The directness of the LVD group included 16 patients, the exclusiveness of LVD included 24 patients, and the venous strain group included 23 patients. In the univariate analysis, risk location (*p* = 0.004), directness of LVD (*p* = 0.019), exclusiveness of LVD (*p* = 0.003), and venous strain (*p* = 0.001) were significantly different in patients with aggressive clinical presentation. In the multivariate analysis, exclusiveness of LVD (*p* = 0.014) and venous strain (*p* = 0.006) were observed, with a significant difference in patients with aggressive clinical presentation compared with those with benign clinical presentation. The concordance statistic measures the goodness of fit for binary outcomes in a logistic regression model. The area under the receiver operating characteristic curve was 0.891 (sensitivity, 83.33%; specificity, 76.92%), which indicated a strong predictive model (Figure 2).

### 3.2. Case Demonstration

Patient 1: A 42-year-old man presented with intermittent headache in the left occipital region for 10 months. He denied related aura-like symptoms, proptosis, chemosis, and bruit. Angiography revealed DAVF in the left posterior temporal region with multiple feeders from the posterior division of the left middle meningeal artery (MMA) and left occipital arteries, directly drained by cortical veins into the left transverse and superior sagittal sinuses without venous ectasia (Figure 3A,B).

Although the patient’s angio-architectural features were compatible with Cognard type III, which is stratified as an angiographically high-risk group, aggressive presentation was not observed in this patient. As suggested by Batsavias et al. [7], the DAVF observed in this patient had a direct LVD and venous strain, and no exclusive LVD, representing a relatively low risk for aggressive clinical presentation. We then planned a stereotactic radiosurgery (SRS), and the DAVF was treated after SRS (Figure 3C,D).

Patient 2: A 46-year-old man presented with a sudden-onset headache and dizziness with right-sided clumsiness. Brain computed tomography revealed left temporal hemorrhage (Figure 4A), and angiography revealed left occipital DAVF fed by the left posterior meningeal artery, left MMA, and left occipital artery with direct LVD into the cortical vein and venous ectasia (Figure 4C,D).

The fistula was classified as Cognard type IV. The DAVF observed in this patient had a direct and exclusive LVD, as well as venous strain (DES), which is classified as a highest-risk DAVF with aggressive clinical presentation. The intracranial hematoma enlarged within 48 h after admission before endovascular treatment (Figure 4B), and emergency craniectomy with clipping of the DAVF was performed. The patient regained consciousness after the surgery, with some sequelae of right hemiparesis.

## 4. Discussion

The cDAVFs are lesions located within the dura, often near the venous sinuses, which account for approximately 10% of all intracranial vascular malformations [9]. Bridging and emissary veins are often related to cDAVFs; however, the exact association between these two is not well described. These cDAVFs may also develop in the bridging and emissary veins [10,11]. The characteristics of cDAVR that contribute to the aggressiveness of clinical behaviors include age [12], sex [13], location of the lesion [14], arterial feeders of the lesion, and venous drainage patterns [7,15]. Conventionally, cDAVFs with LVD are considered to be at high risk of aggressive clinical presentation [9,10,11,12,13,14]. However, venous outflow restrictions due to partial or complete occlusion or stenosis of the involved venous sinus, high-flow shunt, and the risky lesion site also play important roles in the risk of aggressive clinical presentations among patients with cDAVFs [16].

The Borden and Cognard classification systems are the traditional grading systems for cDAVFs [4], which use cortical vein regurgitation to stratify aggressive type from benign type of cDAVFs, so that clinicians can predict the clinical behaviors of patients and decide the treatment strategies. These two grading systems focus on the occurrence of cortical venous regurgitation. However, numerous patients with angiographically aggressive cDAVF have relatively benign clinical presentations, such as pulsatile tinnitus, dizziness, and headache. In contrast, some patients with similar grades present with detrimental clinical presentations, such as dementia, hemorrhage, early rebleeding, and even life-threatening neurological deficits. The previous grading systems need to be revised in terms of the risk of aggressive clinical presentations. For example, none of the above classifications consider the directness and exclusiveness of LVD, cortical venous congestion, and ectasia as risk factors [7] for aggressive clinical presentation. In addition, Satomi et al. proposed that lesion location is also an important predictive factor for aggressive clinical behavior, especially in the absence of cortical venous drainage [14].

In terms of clinical presentations, a series of DAVFs from our institution provides further insight into the risk factors for stratifying high-risk and low-risk groups of aggressive clinical presentations in traditional angiographically aggressive cDAVF, especially the exclusiveness of LVD and venous strain. Shin et al. showed that cDAVFs with isolated venous sinus, direct pial venous drainage, and pseudophlebitic patterns were associated with initially aggressive clinical presentation. Venous ectasia is frequently associated with a hemorrhagic presentation [17]. Chuanhui Li et al. reported that DAVFs with superficial cortical venous drainage, deep venous drainage, or occluded venous sinus are risk factors for intracranial hemorrhage [6]. Occluded venous sinus drainage can contribute to intracranial venous hypertension, which is considered a risk factor for aggressive presentation. Venous strain and ectatic or congested veins (pseudophlebitic appearance) are pathognomonic signs of venous hypertension. Venous hypertension can induce not only cortical edema and seizures but also vein rupture and intracranial hemorrhage [8]. Recently, multiple authors have proposed that venous ectasia is a significant risk factor for hemorrhage and aggressive presentations in DAVFs [8,17]. In this study, 20 of the 23 patients with DAVF with venous ectasia or venous congestion had aggressive clinical presentations (86.9%).

Among the major components of the newly developed grading system proposed by Baltsavias et al. (directness of LVD, exclusiveness of LVD, and venous strain), the latter two components showed more significant roles in the multivariate analysis in our series, although the close association among these three components seem to play more important roles in intracranial venous hypertension, which subsequently induces more aggressive clinical presentations. The importance of venous drainage patterns cannot be overemphasized [6,13]. Therefore, detailed angiographic characteristics should be comprehensively described and evaluated. In the first case presented in the case demonstration section, the patient had no evident exclusiveness of LVD and venous congestion, although he presented with the angiographically aggressive architecture of cDAVF. The patient’s clinical presentation was relatively benign. He received delayed SRS and achieved a good prognosis without any neurological sequelae. In contrast, the patient in the second case, who also had an angiographically aggressive architecture of cDAVF, showed exclusiveness of LVD and evident venous strain. Although the initial clinical presentation was dizziness with small hemorrhage, the subsequent detrimental early rebleeding occurred before the scheduled definite treatment was performed.

In this study, we provided practical indicators in detail to remind the physicians in-charge that some subgroups with high-risk characters should be treated urgently to avoid rapid deterioration, and that some subgroups can be treated conservatively. There are still some methodological limitations in our study. First, although we collected patients with multiple types of aggressive cDAVF, the number of cases for each type was small. Second, this study was conducted using a retrospective approach. In the future, a prospective nationwide study with a large number of patients should be performed.

## 5. Conclusions

Even patients with angiographically aggressive cDAVF based on the Borden and Cognard classifications can be further stratified into the higher and lower risk groups in terms of clinically aggressive presentations. Previous Borden and Cognard classification systems have failed to differentiate such variable ranges for aggressive clinical presentation among patients with angiographically aggressive DAVFs. Further evaluation of the detailed characteristics of LVD, including the presence of directness and exclusivity of the LVD, as well as the induced strain of the leptomeningeal draining veins, provides practical indicators of treatment strategies for cerebral cDAVF. For patients with DAVF with high-risk characteristics of aggressive clinical presentation, urgent surgical or endovascular treatments to treat or downgrade angiographically aggressive DAVFs are strongly recommended.

## Figures and Tables

**Figure 1 jcm-10-05835-f001:**
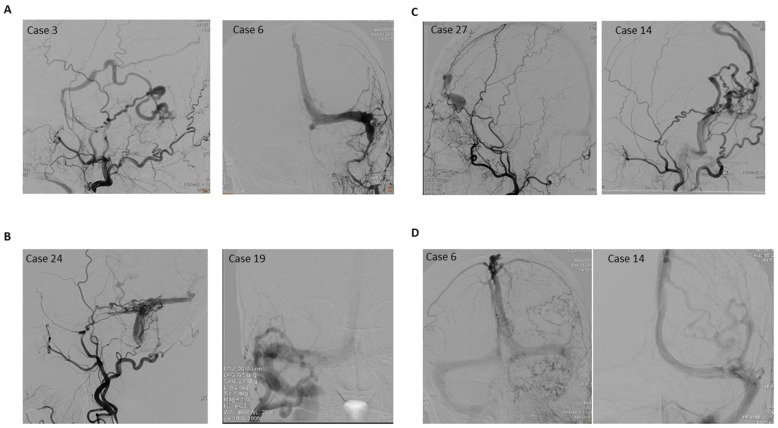
(**A**) **Directness of leptomeningeal venous drainage (LVD):** The intracranial dural arteriovenous fistula (DAVF) in case 3 drains directly into the LVD without sinus involvement, which denotes directness of the LVD (D); the DAVF in case 6 drains into the sinus first, then into the LVD, which denotes non-directness of the LVD (nD). (**B**) **Exclusiveness of LVD:** The DAVF in case 24 drains into an isolated sinus that is not connected with systemic venous drainage, which denotes exclusiveness of LVD (E); the DAVF in case 19 drains into the right sigmoid sinus first, which connects with the systemic venous system. It denotes non-exclusiveness of the LVD (nE). (**C**) **Venous varix of venous strain:** The DAVF in case 27 causes a venous varix at the LVD, which denotes venous strain of the LVD (S); the DAVF in case 14 causes no venous varix, which denotes non-venous strain of LVD (nS). (**D**) **Pseudophlebitis pattern of venous strain:** The venous phase of DAVF in case 6 reveals severe pseudophlebitic pattern, which signifies venous congestion and denotes venous strain of the LVD (S); the venous phase of DAVF in case 14 reveals no evident pseudophlebitic pattern, which denotes non-venous strain of LVD (nS).

**Figure 2 jcm-10-05835-f002:**
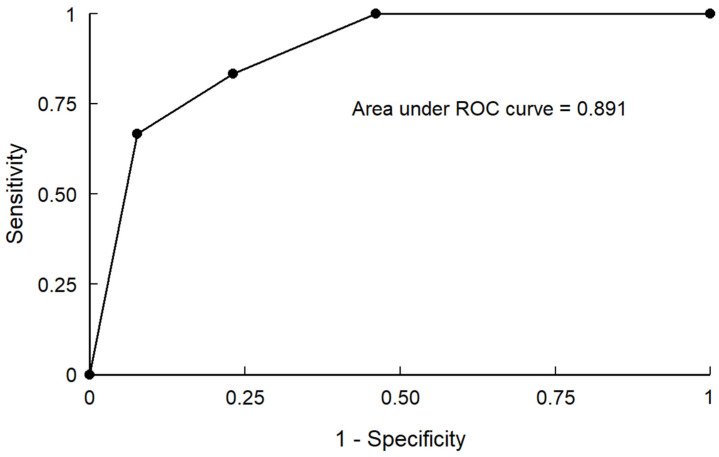
The area under the receiver operating characteristic curve is 0.891, which indicated a strong predictive model.

**Figure 3 jcm-10-05835-f003:**
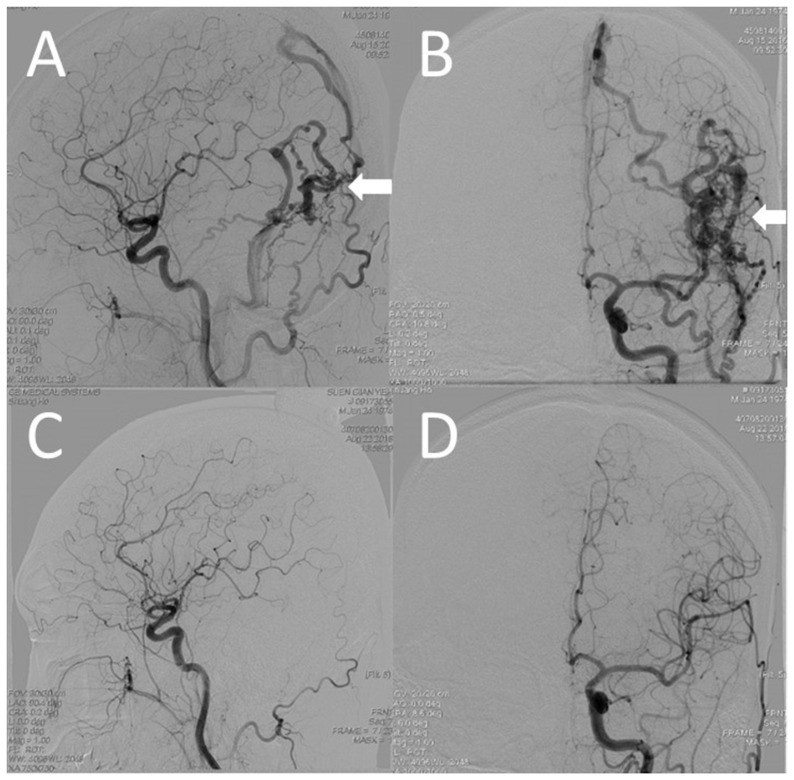
(**A**,**B**) Angiographic images of the left external carotid showing that the dural arteriovenous fistula (DAVF) is supplied through the middle meningeal artery and left occipital arteries, drained by cortical veins into the left transverse and superior sagittal sinuses (white arrow). (**C**,**D**) The DAVF vanished after stereotactic radiosurgery.

**Figure 4 jcm-10-05835-f004:**
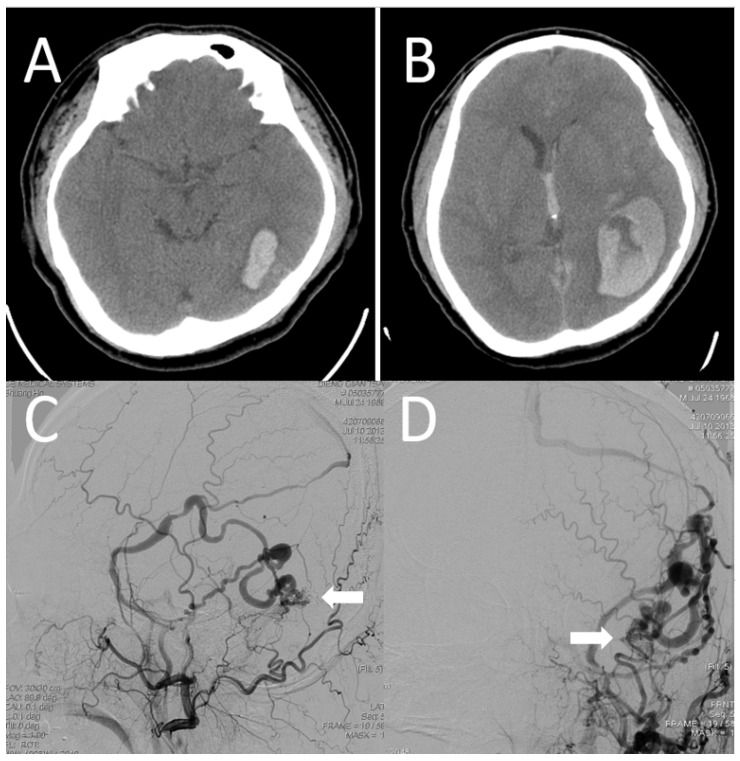
(**A**) A plain computed tomography performed to evaluate severe headache demonstrates a left temporal hematoma; (**B**) the hematoma enlarged within 48 h. (**C**,**D**) Angiography reveals dural arteriovenous fistula with multiple feeders in the left posterior meningeal artery, left middle meningeal artery, and left occipital artery, drained by the leptomeningeal vein only without the venous sinus. Venous ectasia is also observed (white arrow).

**Table 1 jcm-10-05835-t001:** Demographics of studied subjects.

				Univariate Analysis		Multivariant Analysis	
		AP (*n* = 24)	NonAP (*n* = 13)	OR	*p* Value	OR	*p* Value
**Gender**	Male	15	7	1.43	0.609		
	Female	9	6	1			
**Age**	age ≥ 60	13	6	1.38	0.642		
	age < 60	11	7	1			
**Borden Grading**	II	11	9	1	0.179		
	III	13	4	2.66			
**Cognard Grading**	IIa+IIb	11	10		reference		
	III	8	2		0.153		
	IV	5	1		0.199		
**Risk Location**	Yes	18	3	10	**0.004 ***		
	No	6	10	1			
**Directness of LVD**	Yes	14	2	7.7	**0.019 ***		
	No	10	11	1			
**Exclusiveness of LVD**	Yes	20	4	11.3	**0.003 ***	**17.9**	**0.014 ***
	No	4	9	1			
**Venous Strain**	Yes	20	3	16.7	**0.001 ***	**25.4**	**0.006 ***
	No	4	10	1			

AP: Aggressive clinical presentation; NonAP: Nonaggressive clinical presentation; LVD: Leptomeningeal venous drainage; OR: Odds ratio; *: *p* < 0.05.

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
