# Peer review of "Risk Factors of Aggressive Clinical Presentation in Patients with Angiographically Aggressive Cranial Dural Arteriovenous Fistulas"

_jcm, 2021, doi:10.3390/jcm10245835_

Round 1
Reviewer 1 Report
The aim of the study is certainly interesting, namely to explore the potential - additional to the cortical venous reflux - risk factors of aggressive clinical presentations in patients with angiographically aggressive fistulae.
However, the manuscript has several weaknesses and needs major revision.
Please find underlined the points, which need correction or revision and my comments in italics.
Staring from the Abstract, a couple of points:
Compared to nonaggressive cranial dural arteriovenous fistula (cDAVF), aggressive 17 cDAVF often carries leptomeningeal venous drainage (LVD) and has approximately 15% annual 18 risk of hemorrhagic and nonhemorrhagic aggressive neurological presentations. LVD is part of the definition of aggressive CDAVFs, therefore “often” should be deleted
“aggressive cDAVD”… apparently they mean cDAVF
In terms of aggressive clinical presentations, the previous classification could not be insufficiently satisfactory to differentiate the higher risk group from the lower risk group. This sentence is incomprehensible. Apparently they
mean “classifications” in plural. The rest of the sentence should be reformulated. Overall, the manuscript should be edited by a professional English language editing service!
The following sentence in Materials and methods is very confusing: “Direct cortical vein shunting, tentorial area, anterior cranial fossa, cistern magnum, superior sagittal sinus, and petrous venous drainage were considered risky
locations.” This sentence describes risky locations according to the authors and literature.
However:
1. Direct cortical vein shunting is not a location, it does not correspond to any concrete topography or corresponds to many different ones. Besides the authors write next that “All patients were analysed according to the Directness …” Therefore “Direct cortical vein shunting” was included anyway as a factor,
not as a location.
2. “cistern magnum”(which should be corrected to “cisterna magna”) is a cistern with CSF, not a vascular structure. It is unclear what the authors mean. This should be clearly defined.
3. The authors should support with references the idea that SSS is a risky location.
4. “petrous venous drainage” is vague. What does it mean? Drainage to the superior petrosal sinus? Again, this does not represent a location. If they mean petrosal location with drainage through the superior petrosal vein, then it should be written so (suboptimal but acceptable).
The sentence “proposed by Gerasimos Baltsavias et al. [4] should be corrected to “proposed by Baltsavias et al. [4].
In the Table 1, the authors should explain why the Cognard grade IIb was not included (and the “Coganard” should be corrected)
In paragraph 3.1 a similar comment as above. The authors write: “DAVF occurred at a risky location (anterior cranial fossa, tentorium, cistern magnum, bridge vein, and superior sagittal sinus) in 21 patients. “ The “bridge
vein” should be corrected to “bridging vein”. The confusing aspect here is the following: Anterior cranial fossa and tentorium (when correctly analysed) they all have the same angioarchitecture, namely a bridging vein fistula.
Apparently the authors mean bridging vein fistulae in other locations. But then the superior sagittal sinus comes, which is a sinus, not a bridging vein(s). Therefore, once again, the authors need to explain why a sinus (SSS) fistula
is considered a risky location. The same applies for the cisterna magna. If the authors mean a galenic fistula, then they should write it so.
In the fig 2, the authors should put an arrow exactly at the fistula. Was it a posterior temporal bridging vein fistula? If yes, it should be described accordingly.
Page 5, paragraph 3.2: “As suggested by Batsavias et al. [4], the DAVF observed in this patient was direct LMD, not exclusive LVD, and venous strain (duodenal neuroendocrine neoplasm), representing a relatively low risk for
aggressive clinical presentation, was not observed. The sentence “was direct LMD” should be corrected to “had a direct LVD”. Completely unclear what a duodenal neuroendocrine neoplasm has to do with the whole analysis.
Page 5, case 2: “The DAVF observed in this patient was a direct LMD, not an exclusive LVD, and venous strain (DES), which is classified as a high- risk DAVF for aggressive clinical presentation, was observed.” This sentence is full of
mistakes and should be reformulated to “The DAVF observed in this patient had a direct and exclusive LVD, as well as venous strain (DES), which is classified as the highest-risk DAVF for aggressive clinical presentation”
In the conclusion: “Further evaluation of the detailed characteristics of LVD, including the presence of retrograde cortical venous drainage, provides practical indicators of treatment strategies for cerebral aDAVF. “ This is again very confusing. LVD means leptomeningeal venous drainage. This drainage is by definition and always retrograde. Therefore, the above sentence (… valuation of the detailed characteristics of LVD, including the presence of retrograde cortical venous drainage…) represents a tautology. Perhaps the authors (mean?) should write: “…including the presence of directness and exclusivity of the LVD as well as the induced strain of the leptomeningeal draining veins, provides…”
“treatment strategies for cerebral aDAVF “ What letter a stands for?
----------------------------------------------------------------------------
One last comment for the authors, not to be considered as part of their manuscript review: Their reference by Satomi (11) and the results presented in that paper are based on a misconception, despite published in a prestigious journal. In the non-LVR group they included 3 “tentorial CDAVF” cases. A tentorial CDAVF by definition has LVD. Except if by “tentorial” one includes also transverse sinus or superior petrosal sinus or straight sinus (all linked to the tentorium) fistulae. This implies a background problem of uniform definitions.
Author Response
Dear reviewer, Thank you for giving us the opportunity to submit a revised draft of the manuscript “Risk Factors of Aggressive Clinical Presentation in Patients with Angiographically Aggressive Cranial Dural Arteriovenous Fistulas” for publication in the Journal of clinical Medicine. We appreciate the time and effort that you dedicated to providing feedback on our manuscript and are grateful for the insightful comments on and valuable improvements to our paper. We have incorporated most of the suggestions made by the reviewers. Please see below, in red, for a point-bypoint response to the reviewers’ comments and concerns.
The aim of the study is certainly interesting, namely to explore the potential - additional to the cortical venous reflux - risk factors of aggressive clinical presentations in patients with angiographically aggressive fistulae.
Response 1: Thank you
However, the manuscript has several weaknesses and needs major revision.
Please find underlined the points, which need correction or revision and my comments in italics.
Staring from the Abstract, a couple of points:
Compared to nonaggressive cranial dural arteriovenous fistula (cDAVF), aggressive 17 cDAVF often carries leptomeningeal venous drainage (LVD) and has approximately 15% annual 18 risk of hemorrhagic and nonhemorrhagic aggressive neurological presentations. LVD is part of the definition of aggressive CDAVFs, therefore “often” should be deleted “aggressive cDAVD”… apparently they mean cDAVF
Response 2: Thank you for pointing this out, and it has been corrected in the revised manuscript.
In terms of aggressive clinical presentations, the previous classification could not
be insufficiently satisfactory to differentiate the higher risk group from the lower risk group. This sentence is incomprehensible. Apparently they mean “classifications” in plural. The rest of the sentence should be reformulated. Overall, the manuscript should be edited by a professional English language editing service!
Response 3: Thank you for suggestion, and the manuscript was already edited by a professional English language editing service.
The following sentence in Materials and methods is very confusing: “Direct cortical vein shunting, tentorial area, anterior cranial fossa, cistern magnum, superior sagittal sinus, and petrous venous drainage were considered risky
locations.” This sentence describes risky locations according to the authors and literature.
However:
1.Direct cortical vein shunting is not a location, it does not correspond to any concrete topography or corresponds to many different ones. Besides the authors write next that “All patients were analysed according to the Directness …” Therefore “Direct cortical vein shunting” was included anyway as a factor, not as a location.
2.“cistern magnum”(which should be corrected to “cisterna magna”) is a cistern with CSF, not a vascular structure. It is unclear what the authors mean. This should be clearly defined.
Response 4: We agree with the reviewer’s assessment, and it should be foramen magnum.
3. The authors should support with references the idea that SSS is a risky location.
Response 5: Thank you for suggestion, and we have added the references in the revised manuscript.
N. Ohara, S. Toyota, M. Kobayashi, and A. Wakayama. Superior Sagittal Sinus Dural Arteriovenous Fistulas Treated by Stent Placement for an Occluded Sinus and Transarterial Embolization. Interv Neuroradiol. 2012 Sep; 18(3): 333–340.
Cognard C, Gobin YP, Pierot L, et al. Cerebral dural arteriovenous fistulas: clinical and angiographic correlation with a revised classification of venous drainage. Radiology. 1995;194:671–680.
4. “petrous venous drainage” is vague. What does it mean? Drainage to the superior petrosal sinus? Again, this does not represent a location. If they mean petrosal location with drainage through the superior petrosal vein, then it should be written so (suboptimal but acceptable).
Response 6: Yes, it means petrosal location with drainage through the superior petrosal vein.
The sentence “proposed by Gerasimos Baltsavias et al. [4] should be corrected to “proposed by Baltsavias et al. [4].
In the Table 1, the authors should explain why the Cognard grade IIb was not included (and the “Coganard” should be corrected)
Response 7: Thank you for pointing this out, and they have been corrected in the revised manuscript.
In paragraph 3.1 a similar comment as above. The authors write: “DAVF occurred at a risky location (anterior cranial fossa, tentorium, cistern magnum, bridge vein, and superior sagittal sinus) in 21 patients. “ The “bridge vein” should be corrected to “bridging vein”. The confusing aspect here is the following:
Anterior cranial fossa and tentorium (when correctly analysed) they all have the same angioarchitecture, namely a bridging vein fistula.
Apparently the authors mean bridging vein fistulae in other locations. But then the superior sagittal sinus comes, which is a sinus, not a bridging vein(s). Therefore, once again, the authors need to explain why a sinus (SSS) fistula is considered a risky location. The same applies for the cisterna magna. If the authors mean a
galenic fistula, then they should write it so.
Response 8: Thank you, and the change can be found in the revised manuscript.
In the fig 2, the authors should put an arrow exactly at the fistula. Was it a posterior temporal bridging vein fistula? If yes, it should be described accordingly.
Response 9: Thank you for suggestion, and we have put a white arrow in the revised manuscript.
Page 5, paragraph 3.2: “As suggested by Batsavias et al. [4], the DAVF observed in this patient was direct LMD, not exclusive LVD, and venous strain (duodenal neuroendocrine neoplasm), representing a relatively low risk for aggressive clinical presentation, was not observed. The sentence “was direct LMD” should be
corrected to “had a direct LVD”. Completely unclear what a duodenal neuroendocrine neoplasm has to do with the whole analysis.
Page 5, case 2: “The DAVF observed in this patient was a direct LMD, not an exclusive LVD, and venous strain (DES), which is classified as a high- risk DAVF for aggressive clinical presentation, was observed.” This sentence is full of
mistakes and should be reformulated to “The DAVF observed in this patient had a direct and exclusive LVD, as well as venous strain (DES), which is classified as the highest-risk DAVF for aggressive clinical presentation”
In the conclusion: “Further evaluation of the detailed characteristics of LVD, including the presence of retrograde cortical venous drainage, provides practical indicators of treatment strategies for cerebral aDAVF. “ This is again very confusing. LVD means leptomeningeal venous drainage. This drainage is by definition and always retrograde. Therefore, the above sentence (… valuation of the detailed characteristics of LVD, including the presence of retrograde cortical venous drainage…) represents a tautology. Perhaps the authors (mean?) should write: “…including the presence of directness and exclusivity of the LVD as well as the induced strain of the leptomeningeal draining veins, provides…"
“treatment strategies for cerebral aDAVF “ What letter a stands for?
Response 10: We appreciate the reviewer’s feedback, and they have been corrected in the revised manuscript.
----------------------------------------------------------------------------
One last comment for the authors, not to be considered as part of their manuscript review:
Their reference by Satomi (11) and the results presented in that paper are based on a misconception, despite published in a prestigious journal. In the non-LVR group they included 3 “tentorial CDAVF” cases. A tentorial CDAVF by definition has LVD. Except if by “tentorial” one includes also transverse sinus or superior petrosal sinus or straight sinus (all linked to the tentorium) fistulae. This implies a background problem of uniform definitions.
Reviewer 2 Report
The authors aimed to evaluate the angiographical risk factors for differentiating the aggressive presentation from the non-aggressive presentation in a series of patients with angiographical aggressive cranial DAVF. The authors reported that the exclusiveness of leptomeningeal venous drainage and venous strain were significant risk factors for the aggressive clinical presentation in multivariate analysis. The results of this study, which were derived based on the analysis of detailed angiography in DAVFs with LVD, are consistent with previous reports that PPP, venous ectasia, direct pial venous drainage, the exclusiveness of LVD, and cases presenting with isolated venous sinus were associated with an aggressive clinical presentation. However, the results obtained in this study based on venous drainage pattern and venous morphology are not novel.
Major comments:
Abstract
The authors should specify that the patients were of the aggressive type in the conventional classification based on angiographic findings (Borden II and III, Cognard type IIa+b to IV).
Introduction
There is no explanation of angiographical aggressive DAVF (page 2, line 58), and the clarification of the subject seemed unclear in the introduction.
Materials and Methods
The reason for choosing the location specified as risky location (page 2, line 78) should be clearly stated. In addition, it is better to cite the references.
The factors employed as covariates in the logistic regression analysis should be clearly indicated. The conditions under which the covariate factor was employed (e.g., P<0.2 in univariate analysis) should also be provided. Depending on the number of factors used as covariates, the sample size of this study may not be sufficient.
The author analyzed the potential risk factors such as directness of LVD, exclusiveness of LVD, and venous strain. These unfamiliar terms are explained in the materials and methods with appropriate references, but honestly it is difficult to understand. It would be better to provide illustrations of typical angiographic findings that reflect these factors, as well as the schemes that show them.
Results
In the Demographics of patients, the location of DAVF was presented in Table 1 as risk location (yes or no), but the distribution of the locations in the study was unclear. The clear shunt locations and their numbers should be provided.
In the AUC obtained from the ROC analysis, I don't know what factors are associated with aggressive clinical presentation. Which factor was used for the ROC analysis for predicting the aggressive clinical presentation? Did you use the exclusiveness of LVD or venous strain that were significant in multivariate analysis? Specificity, sensitivity, positive predictive value and negative predictive value should be calculated.
Case demonstration
Patient 1 presented non-aggressive clinical presentation with angiographical aggressive DAVF. Among the angiographic factors specified by the author, direct LMD was present, but exclusive LVD and venous strain were not present in this case. Am I understanding you correctly?
Patient 2 presented aggressive clinical presentation (intracerebral hemorrhage) with angiographical aggressive DAVF. The author described that DAVF observed in this patient was a direct LMD and venous strain, but an exclusive LVD was not observed (page 6, line 15: not an exclusive LVD). Am I understanding you correctly?
I think that Patient 2 is inappropriate case for the case to reflect the results of the multivariate analysis that the exclusiveness of LVD and venous strain were risk factors. In addition, this case was a case of rebleeding after 48 hours. This case does not reflect the comparative medium-term follow-up of the patient with angiographical risk factor of aggressive clinical presentation.
Figure 3
The author showed venous ectasia with white arrow in Figure 3C and D. Were the locations of white arrows appropriate?
Discussion
The authors should mention a discussion of why direct LVD was not a risk factor for aggressive clinical presentation.
Round 2
Reviewer 2 Report
The author analyzed the potential risk factors such as directness of LVD, exclusiveness of LVD, and venous strain. These unfamiliar terms are explained in the materials and methods with appropriate references, but honestly it is difficult to understand. I still found it difficult to understand the key concept of this study. It would be better for readers' understanding if each element was clearly explained with illustrations, as recommended previously.
Author Response
Thank you for suggestion, and we have added the illustrations (Figure 1) in the revised manuscript.